

# Physical characteristics of elite youth male football players aged 13–15 are based upon biological maturity

Shidong Yang[1,2] and Haichun Chen[1]

[1] School of Physical Education and Sport Science, Fujian Normal University, Fuzhou, Fujian, China
[2] Department of Physical Education, Nanjing Xiaozhuang University, Nanjing, Jiangsu, China

Corresponding author
Haichun Chen, chc1129@163.com

## ABSTRACT

**Background:** Older and more mature football players have been reported to gain advantages in the selection process during adolescence. The aim of this study was to investigate the influence of skeletal age (SA) on the physical characteristics of elite male football players aged 13–15 years through a cross-sectional study.

**Methods and Materials:** We enrolled 167 elite players aged 13–15 from three academic football schools in China, and measured height, body mass, thigh circumference, skinfold (triceps and calf), 10 m/30-m sprint, $T$-tests (left and right), 5 × 25-m repeated-sprint ability (5 × 25 RSA), standing long jump, and YoYo intermittent recovery test level 1 (YYIR1). Subjects were divided into early-, average-, and late-maturity levels according to their SA and chronological age (CA) based on the following criteria: SA-CA > +1 year, SA-CA = ±1 year, and SA-CA < −1 year, respectively. The differences in parameters among the groups were analyzed using one-way analysis of variance and Bonferroni's *post-hoc* test, with statistical significance set at $p < 0.05$.

**Results:** Relative to the late-maturing players, the early- and average-maturing players aged 13–15 years were taller, heavier, had a larger thigh circumference, and scored higher on the standing long jump, 30 m sprint, and 5 × 25-RSA ($p < 0.05$). The physical (except for body-fat percentage) and athletic characteristics of players aged 13–15 were not only significantly influenced by biological maturity, but also increased significantly with CA. The influence of biological maturity on height, 30-m sprints, and 5 × 25-m RSA diminished with age and exerted no significant effect on body-fat percentage and on YYIR1. Late-maturing players exhibited the greatest increase in physical (except for body-fat percentage) and athletic performance (except for the 10-m sprint) compared to players of early and/or average maturity.

**Conclusions:** Although early-maturing players aged 13–15 possessed better anthropometric and physical performance than late- and average-maturing players, the growth and development of physical function of late- and average-maturing players was significantly greater, particularly with respect to height, sprint speed, and muscular power.

## INTRODUCTION

Research on the physical performance characteristics of youth football players has made significant progress in recent years. Studies have revealed that elite youth football players are superior in anthropometrics (height, body mass, and body composition) (*Bidaurrazaga-Letona et al., 2016*; *Deprez et al., 2015a*; *Nughes et al., 2020*; *Bongiovanni et al., 2020*) and physical characteristics (speed, agility, anaerobic endurance, strength, and power) (*Emmonds et al., 2016*; *Aquino et al., 2017*; *Grendstad et al., 2020*; *Deprez et al., 2015b*), which are important factors influencing selection by coaches (*Patel et al., 2020*; *Bidaurrazaga-Letona et al., 2019*; *Nughes et al., 2020*). As such, football players' physical characteristics significantly influence the selection of elite *vs* non-elite players while they are still youths (*Itoh & Hirose, 2020*).

Researchers have demonstrated the influence of maturation status on physical characteristics of youth players (*Altimari et al., 2018*; *Grendstad et al., 2020*; *Fragoso, Massuca & Ferreira, 2015*). Biological maturity is an important factor that affects the physical indices of elite youth football players relative to non-elite players. Teenagers with early, average, and late biological maturation are seen in the same chronological age (CA) group in many football organizations worldwide, including school teams and football clubs. If coaches are to use the physical test data to make informed decisions regarding the "athleticism" of youth male football players, so as to inform the design of training plans, then coaches need to be aware of the impact maturity may have on the development of specific types of physical fitness.

In young footballers, secondary sexual maturation (*Malina et al., 2004*), skeletal age (SA) (*Tanner, 1959*; *Roche, Eyman & Davila, 1971*), chronological age at peak velocity (PHV) (*Philippaerts et al., 2006*; *Mirwald et al., 2002*), and testicular volume (*Young et al., 1968*) have been previously employed to determine biological maturity. However, these assessment methods have limitations. For example, evaluating secondary sexual maturation and measuring testicular volume are not always acceptable to players, parents, or clubs (*Malina et al., 2004*).

Currently, SA is the most accurate and commonly used method of evaluating biological maturity worldwide (*Itoh & Hirose, 2020*; *Eston & Reilly, 2009*), and Tanner-Whitehouse (TW, *Tanner, 1959*; *Poznanski, 1977*), Greulich-Pyle (GP, *Garn & Lewis, 1959*), and Fels (*Chumela, Roche & Thissen, 1989*) are common methods of estimating SA. These methods are similar to each other in principle. Hand-wrist X-rays of children and adolescents are compared using a set of criteria; however, signs of bone maturity and the criteria specific to each method differ (*Vignolo et al., 1992*).

The TW method is used to evaluate SA based on the maturational indicators of 20 bones: the radius, ulna; 11 metacarpals; the phalanges of the first, third, and fifth digital rays; and seven carpals, excluding the pisiform (*Tanner, 1959*). The latest revised TW3 is widely used internationally to assess the biological maturity of children and adolescents (*Eston & Reilly, 2009*; *Hochberg, 2016*; *Altimari et al., 2018*; *Hsieh et al., 2013*), and it can be used as a standard index of biological maturity of youth football players (*Malina et al., 2018*). Although the TW method has been revised twice, with the second revision (TW3)

retaining the radius-ulna-short SA (RUS SA) and carpal SA from the first revision (TW2), the 20-Bone SA was eliminated (*Tanner, Goldstein & Cameron, 2001*). The TW3 method does not include modified criteria with regard to maturity indicators, and it has been used to assign scores for each bone. However, it was converted to the sum of the maturity scores for the radius, ulna, and short bones (RUS maturity scores) into a modified SA.

Although many studies have shown that SA has a positive correlation with physical indices of youth male football players (*Malina et al., 2004*; *Selmi et al., 2020*; *Rommers et al., 2019*, *Duarte et al., 2019*; *McCunn et al., 2017*), the effects of SA on physical characteristics have been studied separately; the data have suggested this is appropriate. For example, *Bongiovanni et al. (2021)* suggested that the arm muscle area and circumference and suprapatellar girths are closely related to sprint- and intermittent-aerobic endurance in elite youth football players. In addition, few investigators have reported differences in the growth and development of the physical characteristics of young male footballers at different stages of biological maturity (*i.e.*, early-, average-, and late-maturity).

In this cross-sectional study, we determined the effects of biological maturity on the physical characteristics of elite male football players 13–15 years of age. We also investigated player growth and development with respect to these endpoints in these specific age groups of football players at distinct levels of maturity. Such findings will assist strength and conditioning coaches to better understand the influence of maturation on the development of physical characteristics of 13- to 15-year-old male football players. We hypothesized that early-maturing players would possess a larger physique—related to height, body mass, and thigh circumference—compared to late-maturing players, with the physical growth of late-maturing players more likely to be affected by biological maturity. In addition, we speculated that early-maturing players would exhibit superior physical abilities—such as better standing long jump, sprint times, and endurance—than late-maturing players. Finally, we posited that height and body mass would be positively correlated with jumping ability, in addition to a negative correlation with sprint and speed endurance.

## MATERIALS AND METHODS

We investigated skeletal maturity between September 26 and October 7 of 2020. The anthropometric and field tests were performed during the fourth weekend of September and during the first and second week of October prior to the start of the 2020–2021 football season. There were no high-intensity competitions or training for 48 h prior to the test, and all subjects were free of injury at the time of study. We conducted all measurements and evaluations as part of a team's selection program.

### Participants

A total of 167 male football players (59 13-year-olds, 56 14-year-old-olds, and 52 15-year-olds) were recruited from three football schools in China. Goalkeepers and foreign football players were excluded from our study. Age categories were defined by chronological age as of October 1, 2020. All advantages and risks of the investigation were

explained to all participants, coaches, and players' legal guardians; and written informed consent was obtained. Subjects were permitted to withdraw in the case of injury or simply if they chose not to take the field test. This study was implemented according to the guidelines of the Helsinki Declaration and conducted with the approval of the Fujian Normal University Ethics Committee on Human Experimentation.

## Protocols

We measured height, body mass, skinfold thickness, and thigh circumference to evaluate the physical characteristics of players aged 13–15. We measured sprinting, jumping, agility, and aerobic performance to evaluate the physical characteristics of the same players. All anthropometric measurements were conducted at 8:00 AM, and height and body mass were measured once. The circumference of the thigh and the skinfold at the triceps and calf were each measured twice. Our calculated intraclass correlation coefficients (ICC) were evaluated as "almost perfect" (ICC > 0.75) according to a previous study (*McGraw & Wong, 1996*).

The test battery was performed following the guidelines of the China Football Association, to evaluate physical abilities such as speed, agility, jump length, and endurance that are required for football. We measured 10-m and 30-m sprints, *T*-tests, standing long jump (SLJ), $5 \times 25$-m repeated sprints, and Yo–Yo intermittent recovery test level 1 (YYIR1).

Participants sequentially performed 10-m and 30-m sprints, SLJ, *T*-tests, and YYIR1. YYIR1 was conducted after completing the other tests; and the $5 \times 25$-m RSA was tested at the same time on the next day. All physical measurements were executed twice with a rest period of 3 min or more between trials, except for the $5 \times 25$-m repeated sprints and YYIR1, which were each performed once. Before the test, players warmed up with 10 min of jogging and dynamic movements, and 5 min of sprinting and jumping with progressive intensity.

## Procedures

### Classification

According to the TW3 method, SA was evaluated based on X-rays of the players' left wrists (*Carty, 2002*). Players were divided into three maturity categories within each CA group: (1) early, SA-CA > +1 year; (2) average, SA-CA = ±1 year; and (3) late, SA-CA < −1 year (*Malina et al., 2018*). Participants and the evaluator did not know the groupings, and all measurements were performed in a double-blinded manner.

## Anthropometric and skinfold measurements

Physical characteristics were measured with the participants barefoot and wearing pants and a T-shirt. Height was measured with a YL–65S stadiometer (Yagami, Nagoya, Japan) to the nearest 0.1 cm, body mass was measured with a XiaoMi body-fat monitor (XiaoMi, Beijing, China) to the nearest 0.1 kg, and skinfold thickness was determined with a Harpenden skinfold caliper (system1; Skyndex, Caldwell, NC, USA) to the nearest 0.01 mm. The exact positioning of each skinfold measurement was in accordance with

procedures described previously (*Norton et al., 2000*). All skinfold measurements were taken on the right side of the body, and body-fat percentage (% Fat) was calculated from skinfold thicknesses (triceps and calf) using Slaughter's equations (*Slaughter et al., 1988*). The intra-rater reliability ICCs (3, 1) of the skinfold thicknesses of the triceps and calf were 0.88 and 0.92, respectively.

## Circumference measurement

The thigh circumference of the right leg was measured with a standard tape measure to the nearest 0.1 cm. During measurement, the tape measure was pulled to avoid twisting and over-tightening around the thigh. We first measured the circumference of the thigh from a point 5 cm proximal to the upper end of the patella (*Mathur et al., 2008*), and then measured the circumference of the thigh to obtain the average. The intra-rater reliability ICC (3, 1) for thigh circumference was 0.9.

## Sprint and agility measurements

Sprint times were assessed over 10 m and 30 m using timing gates (Brower Timing System, Draper, UT, USA). Subjects started 0.5 m behind the first timing gate and ran maximally past the 30-m timing gate. Times were recorded to the nearest 0.01 s with the quicker of the two attempts used for the sprint score. The intra-rater reliability ICCs (3, 1) for the 10-m sprint and 30-m sprint tests were 0.9 and 0.9, respectively.

Agility was evaluated with *T*-tests (*Vandendriessche et al., 2012*). In this test, players ran 5 m straight from the starting line, turned 90° and ran 5 m before the next turn of 180°, ran 10 m toward the second 180° turn, and then ran 5 m towards the last turn of 90°—ultimately finishing at the starting line. The first *T*-test was executed with all turns performed to the left, and the second *T*-test with all turns performed to the right. Times were recorded to the nearest 0.01 s.

## Jump measurement

The standing long jump was carried out to assess the players' explosive power. All football players placed both feet behind the starting line and jumped as far as possible, while landing on both feet. The distance from the line to the player's closest heel was then measured with a measuring tape. The test was performed twice, and the longest jump distance between the two measurements was used for the analysis. The intra-rater reliability ICC (3, 1) for the standing long jump tests was 0.86.

## Aerobic capacity measurement

The YYIR1 was used to estimate aerobic capacity. *Krustrup et al. (2003)* reported YYIR1 to be a valid and reliable test in assessing specific fitness for football. During the test, the players performed a series of 20-m runs—resting for 5 s every 40 m—and then ran with progressive increments in speed and for increasingly shorter time periods between changes. Failure to achieve the shuttle run in time on two occasions resulted in termination of the test, and we then recorded the total running distance.

## Speed and endurance measurements

The 5 × 25-m repeated sprint was used to evaluate anaerobic endurance. On a 25-m straight line, marker barrels were placed at intervals of 5 m. On hearing the command to "run", the participants ran from the starting line to the first marker barrel and knocked it down with their hands, then turned back to the starting line and again knocked down the marker barrel with their hands; this task was repeated for the second and remaining marker barrels. The runners sequentially knocked down all the marker barrels, and ultimately sprinted back to the starting line. Times were recorded to the nearest 0.01 s.

## Statistical analyses

Measurement results are expressed as means ± SD. The Kolmogorov–Smirnov test and Levene's test were used to assess normality and homogeneity of all test data, respectively. One- way analysis of variance was used to analyze the differences in each age-measurement parameter among the early-, average-, and late-maturing groups; and Bonferroni's post-hoc test was performed to determine which measurements were significantly different. The effect size (ES) $\eta^2$ was reported in the parametric data analysis using Eta-squared (*Khalilzadeh & Tasci, 2017*). Eta squared ($\eta^2$) between 0.01 and 0.06 was considered to constitute a small effect; 0.06–0.14, medium; and 0.14 and greater, a large effect (*Cohen, 1988*). The Pearson correlation coefficient was used to investigate the correlation between each anthropometric index and each physical ability, with $r > 0.5$ regarded as a strong correlation (*Thomas, Nelson & Silverman, 2015*). We also used the least-squares linear regression equation ($y = a + bx$; b = slope) to compute the slopes between CA and the physical parameters of early-, average- and late-maturing groups to determine the influence of biological maturity on the physical characteristics of players aged 13–15 years. Statistical significance was set at 0.05, and all data were analyzed using SPSS software (version 26; IBM, Armonk, NY, USA).

## RESULTS

In Table 1 we summarize the descriptive statistics for the physical characteristics of the players by early-, average-, and late maturation. We observed no significant differences in body-fat percentage (% Fat) or aerobic endurance among early-, average-, and late-maturing players from 13 to 15 years of age. There were, however, significant differences in height (ES = 0.527, $p < 0.001$; ES = 0.39, $p < 0.001$; ES = 0.24, $p < 0.001$, respectively), body mass (ES = 0.221, $p = 0.001$; ES = 0.372, $p < 0.001$; ES = 0.308, $p < 0.001$, respectively), and standing long jump (ES = 0.346, $p < 0.001$; ES = 0.337, $p < 0.001$; ES = 0.446, $p < 0.001$, respectively) among these three groups.

The developmental characteristics with respect to physical capacities of 13- to 15-year-old players with early, average, and late maturation are depicted in Table 2. The body mass (ES = 0.338, $p = 0.03$), thigh circumference (ES = 0.295, $p = 0.032$), 5 × 25-m repeated sprints (ES = 0.337, $p = 0.001$), *T*-test left (ES = 0.289, $p = 0.004$), *T*-test right (ES = 0.328, $p = 0.001$), standing long jump (ES = 0.239, $p = 0.011$), and YYIR1 (ES = 0.163, $p = 0.053$) of 13–15-year-old players showed that these indices in early-maturing players increased significantly; but that height, % Fat and 10-m and 30-m sprint speed did not increase

**Table 1 Comparison of anthropometric and physiological characteristics of early-, average-, and late-maturing players in various age groups from 13 to 15 years old.** The comparison of anthropometric and physiological abilities of early-, average-, and late-maturity football players in each age group from 13 to 15 years old, and the calculated effect size.

| Age | | Early | Average | Late | df | F | p | $\eta^2$ |
|---|---|---|---|---|---|---|---|---|
| 13 (y) | N = 59 | 15 | 33 | 11 | 2 | | | |
| | Height (cm) | $173.6 \pm 3.77^{\dagger,\ddagger}$ | $163.8 \pm 4.93^{\dagger,\S}$ | $159.6 \pm 5.83^{\ddagger,\S}$ | | 31.194 | <0.001 | 0.527 |
| | Body mass (kg) | $55.5 \pm 6.63^{\dagger,\ddagger}$ | $51.1 \pm 4.94^{\dagger}$ | $47.0 \pm 5.31^{\ddagger}$ | | 7.921 | 0.001 | 0.221 |
| | C thigh (cm) | $50.1 \pm 3.79\ddagger$ | $48.9 \pm 3.28^{\S}$ | $43.71 \pm 3.81^{\ddagger,\S}$ | | 11.819 | 0.001 | 0.297 |
| | % Fat | $11.49 \pm 1.53$ | $12.27 \pm 1.87$ | $12.03 \pm 2.22$ | | 0.917 | 0.426 | 0.032 |
| | 5 × 25-m RSA (s) | $35.29 \pm 0.93^{\dagger,\ddagger}$ | $36.15 \pm 0.87^{\dagger,\S}$ | $37.76 \pm 1.15^{\ddagger,\S}$ | | 21.995 | <0.001 | 0.440 |
| | Left T (s) | $8.49 \pm 0.35^{\ddagger}$ | $8.77 \pm 0.45^{\S}$ | $9.18 \pm 0.39^{\ddagger,\S}$ | | 8.822 | <0.001 | 0.240 |
| | Right T (s) | $8.58 \pm 0.40^{\ddagger}$ | $8.79 \pm 0.45^{\S}$ | $9.20 \pm 0.39^{\ddagger,\S}$ | | 6.808 | 0.001 | 0.196 |
| | SLJ (m) | $221.3 \pm 10.40^{\dagger,\ddagger}$ | $209.9 \pm 16.21^{\dagger,\S}$ | $191.5 \pm 9.05^{\ddagger,\S}$ | | 14.815 | <0.001 | 0.346 |
| | 10 m (s) | $1.84 \pm 0.13^{\ddagger}$ | $1.94 \pm 0.19^{\S}$ | $2.11 \pm 0.13^{\ddagger,\S}$ | | 7.862 | 0.001 | 0.219 |
| | 30 m (s) | $4.48 \pm 0.16^{\dagger,\ddagger}$ | $4.77 \pm 0.27\dagger^{\S}$ | $5.13 \pm 0.17^{\ddagger,\S}$ | | 24.977 | <0.001 | 0.471 |
| | YYIR1 (m) | $1{,}962.7 \pm 197.3$ | $1{,}871.5 \pm 257.6$ | $1{,}816.4 \pm 256.3$ | | 1.249 | 0.295 | 0.043 |
| 14 (y) | N = 56 | 10 | 31 | 15 | | | | |
| | Height (cm) | $176.4 \pm 4.01^{\ddagger}$ | $171.9 \pm 5.13^{\S}$ | $164.6 \pm 6.13^{\ddagger,\S}$ | | 16.945 | <0.001 | 0.390 |
| | Body mass (kg) | $62.6 \pm 4.67^{\dagger,\ddagger}$ | $55.9 \pm 5.34^{\dagger,\S}$ | $50.2 \pm 6.17^{\ddagger,\S}$ | | 15.726 | <0.001 | 0.372 |
| | C thigh (cm) | $54.0 \pm 1.93$ | $51.4 \pm 6.44$ | $49.2 \pm 3.81$ | | 2.461 | 0.095 | 0.085 |
| | % Fat | $11.81 \pm 0.97$ | $11.63 \pm 1.23$ | $11.08 \pm 1.06$ | | 1.540 | 0.224 | 0.055 |
| | 5 × 25-m RSA (s) | $33.96 \pm 1.48^{\ddagger}$ | $35.02 \pm 1.58$ | $36.17 \pm 2.12^{\ddagger}$ | | 5.086 | 0.010 | 0.161 |
| | Left T (s) | $8.21 \pm 0.28^{\ddagger}$ | $8.42 \pm 0.41$ | $8.73 \pm 0.72^{\ddagger}$ | | 3.604 | 0.034 | 0.120 |
| | Right T (s) | $8.22 \pm 0.29^{\ddagger}$ | $8.44 \pm 0.41$ | $8.72 \pm 0.66^{\ddagger}$ | | 3.531 | 0.036 | 0.118 |
| | SLJ (m) | $233.1 \pm 11.89^{\ddagger}$ | $224.0 \pm 12.18^{\S}$ | $208.2 \pm 13.52^{\ddagger,\S}$ | | 13.461 | <0.001 | 0.337 |
| | 10 m (s) | $1.74 \pm 0.17$ | $1.88 \pm 0.20$ | $1.88 \pm 0.16$ | | 2.297 | 0.110 | 0.080 |
| | 30 m (s) | $4.39 \pm 0.25^{\ddagger}$ | $4.61 \pm 0.30$ | $4.74 \pm 0.20^{\ddagger}$ | | 5.064 | 0.010 | 0.160 |
| | YYIR1 (m) | $2{,}052.0 \pm 66.1$ | $2{,}047.1 \pm 162.1$ | $2{,}002.7 \pm 134.6$ | | 0.565 | 0.572 | 0.021 |
| 15 (y) | N = 52 | 11 | 31 | 10 | | | | |
| | Height (cm) | $177.2 \pm 5.65^{\ddagger}$ | $175.1 \pm 5.58^{\S}$ | $168.2 \pm 5.86^{\ddagger,\S}$ | | 7.755 | 0.001 | 0.240 |
| | Body mass (kg) | $64.4 \pm 5.64^{\ddagger}$ | $61.0 \pm 5.02^{\S}$ | $54.1 \pm 4.94^{\ddagger,\S}$ | | 10.887 | <0.001 | 0.308 |
| | C thigh (cm) | $54.1 \pm 3.04$ | $53.3 \pm 4.08$ | $50.4 \pm 3.04$ | | 3.216 | 0.310 | 0.116 |
| | % Fat | $12.36 \pm 1.33$ | $12.41 \pm 1.66$ | $11.36 \pm 1.54$ | | 1.739 | 0.186 | 0.066 |
| | 5 × 25-m RSA (s) | $33.51 \pm 1.13^{\ddagger}$ | $34.08 \pm 1.31$ | $34.99 \pm 0.99^{\ddagger}$ | | 3.944 | 0.026 | 0.139 |
| | Left T (s) | $8.04 \pm 0.31^{\dagger,\ddagger}$ | $8.39 \pm 0.35^{\dagger}$ | $8.65 \pm 0.55^{\ddagger}$ | | 6.706 | 0.003 | 0.215 |
| | Right T (s) | $8.05 \pm 0.31^{\ddagger}$ | $8.38 \pm 0.41$ | $8.66 \pm 0.55^{\ddagger}$ | | 5.535 | 0.007 | 0.184 |
| | SLJ (m) | $236.1 \pm 15.02^{\ddagger}$ | $239.9 \pm 12.09^{\S}$ | $213.2 \pm 3.74^{\ddagger,\S}$ | | 19.759 | <0.001 | 0.446 |
| | 10 m (s) | $1.81 \pm 0.19$ | $1.85 \pm 0.18$ | $1.87 \pm 0.018$ | | 0.318 | 0.729 | 0.013 |
| | 30 m (s) | $4.43 \pm 0.19$ | $4.44 \pm 0.26$ | $4.62 \pm 0.21$ | | 2.541 | 0.089 | 0.094 |
| | YYIR1 (m) | $2{,}105.5 \pm 104.3$ | $2{,}035.5 \pm 176.5$ | $2{,}006.0 \pm 77.2$ | | 1.300 | 0.282 | 0.050 |

Notes:
SLJ, standing long jump; 5 × 25-m RSA, 5 × 25-m repeated sprint ability; YYIR1, YoYo intermittent recovery test level 1.
[†] Significant difference between the early- and average-maturing groups: $p < 0.05$.
[‡] Significant difference between the early- and late-maturing groups: $p < 0.05$.
[§] Significant difference between the average- and late-maturing groups: $p < 0.05$.

**Table 2 Growth and development of early-, average-, and late-maturing groups of 13–15-year-old players.** The growth and development degree of early-, average-, and late-maturing groups of 13–15 years old players, and the calculated effect size.

| Age | | 13 (y) | 14 (y) | 15 (y) | df | F | p | $\eta^2$ |
|---|---|---|---|---|---|---|---|---|
| Early | N = 36 | 15 | 10 | 11 | 2 | | | |
| | Height (cm) | 173.6 ± 3.77 | 176.4 ± 4.01 | 177.2 ± 5.65 | | 2.371 | 0.109 | 0.126 |
| | Body mass (kg) | 55.5 ± 6.63[†,‡] | 62.6 ± 4.67[†] | 64.4 ± 5.64[‡] | | 8.412 | 0.001 | 0.338 |
| | C thigh (cm) | 50.1 ± 3.79[†,‡] | 54.0 ± 1.93[†] | 54.1 ± 3.04[‡] | | 6.896 | 0.003 | 0.295 |
| | % Fat | 11.49 ± 1.53 | 11.81 ± 0.97 | 12.36 ± 1.33 | | 1.337 | 0.276 | 0.075 |
| | 5 × 25-m RSA (s) | 35.29 ± 0.93[†,‡] | 33.96 ± 1.48[†] | 33.51 ± 1.13[‡] | | 8.374 | 0.001 | 0.337 |
| | Left T (s) | 8.49 ± 0.35[‡] | 8.21 ± 0.28 | 8.04 ± 0.31[‡] | | 6.718 | 0.004 | 0.289 |
| | Right T (s) | 8.58 ± 0.40[‡] | 8.22 ± 0.29 | 8.05 ± 0.31[‡] | | 8.061 | 0.001 | 0.328 |
| | SLJ (m) | 221.3 ± 10.49[‡] | 233.1 ± 12.0 | 236.0 ± 14.95[‡] | | 5.196 | 0.011 | 0.239 |
| | 10 m (s) | 1.84 ± 0.13 | 1.74 ± 0.17 | 1.81 ± 0.19 | | 1.307 | 0.284 | 0.073 |
| | 30 m (s) | 4.48 ± 0.16 | 4.39 ± 0.25 | 4.43 ± 0.19 | | 0.608 | 0.550 | 0.036 |
| | YYIR1 (m) | 1,962.7 ± 197.2 | 2,052.0 ± 66.1 | 2,105.5 ± 104.3 | | 3.221 | 0.053 | 0.163 |
| Average | N = 95 | 33 | 31 | 31 | | | | |
| | Height (cm) | 163.8 ± 4.93[†,‡] | 171.9 ± 5.13[†,§] | 175.1 ± 5.58[‡,§] | | 40.541 | <0.001 | 0.468 |
| | Body mass (kg) | 51.1 ± 4.93[†,‡] | 55.9 ± 5.34[†,§] | 61.0 ± 5.02[‡,§] | | 29.811 | <0.001 | 0.393 |
| | C thigh (cm) | 48.9 ± 3.28[‡] | 51.4 ± 6.44 | 53.3 ± 4.08[‡] | | 7.040 | 0.001 | 0.133 |
| | % Fat | 12.27 ± 1.87 | 11.63 ± 1.23 | 12.41 ± 1.66 | | 2.070 | 0.141 | 0.043 |
| | 5 × 25-m RSA (s) | 36.15 ± 0.87[†,‡] | 35.02 ± 1.58[†,§] | 34.08 ± 1.31[‡,§] | | 20.874 | <0.001 | 0.312 |
| | Left T (s) | 8.77 ± 0.45[†,‡] | 8.42 ± 0.41[†] | 8.39 ± 0.35[‡] | | 8.515 | <0.001 | 0.156 |
| | Right T (s) | 8.79 ± 0.45[†,‡] | 8.44 ± 0.41[†] | 8.38 ± 0.41[‡] | | 8.749 | <0.001 | 0.160 |
| | SLJ (m) | 209.9 ± 16.21[†,‡] | 224.0 ± 12.18[†,§] | 239.9 ± 12.10[‡,§] | | 38.500 | <0.001 | 0.456 |
| | 10 m (s) | 1.94 ± 0.19 | 1.88 ± 0.20 | 1.85 ± 0.18 | | 2.040 | 0.140 | 0.042 |
| | 30 m (s) | 4.77 ± 0.27[‡] | 4.61 ± 0.30 | 4.44 ± 0.26[‡] | | 11.390 | <0.001 | 0.198 |
| | YYIR1 (m) | 1,871.5 ± 257.6[†,‡] | 2,047.1 ± 162.1[†] | 2,035.5 ± 176.5[‡] | | 7.450 | 0.001 | 0.139 |
| Late | N = 36 | 11 | 15 | 10 | | | | |
| | Height (cm) | 159.6 ± 5.83[‡] | 164.6 ± 6.13 | 168.2 ± 5.86[‡] | | 5.467 | 0.009 | 0.249 |
| | Body mass (kg) | 47.0 ± 5.32[‡] | 50.2 ± 6.17 | 54.1 ± 4.94[‡] | | 4.279 | 0.023 | 0.206 |
| | C thigh (cm) | 43.7 ± 3.81[†,‡] | 49.3 ± 3.81[†] | 50.4 ± 2.17[‡] | | 12.012 | <0.001 | 0.421 |
| | % Fat | 12.03 ± 2.22 | 11.08 ± 1.06 | 11.36 ± 1.54 | | 1.109 | 0.356 | 0.063 |
| | 5 × 25-m RSA (s) | 37.76 ± 1.15[‡] | 36.17 ± 2.12 | 34.99 ± 0.99[‡] | | 7.944 | 0.002 | 0.325 |
| | Left T (s) | 9.18 ± 0.39 | 8.73 ± 0.72 | 8.65 ± 0.55 | | 2.691 | 0.081 | 0.140 |
| | Right T (s) | 9.20 ± 0.39 | 8.72 ± 0.66 | 8.66 ± 0.54 | | 3.163 | 0.055 | 0.161 |
| | SLJ (m) | 191.5 ± 9.05[†,‡] | 208.2 ± 13.36[†] | 213.2 ± 3.77[‡] | | 13.576 | <0.001 | 0.451 |
| | 10 m (s) | 2.11 ± 0.13[‡] | 1.88 ± 0.16 | 1.87 ± 0.18[‡] | | 8.193 | 0.001 | 0.332 |
| | 30 m (s) | 5.13 ± 0.17[†,‡] | 4.74 ± 0.20[†] | 4.62 ± 0.21[‡] | | 19.240 | <0.001 | 0.538 |
| | YYIR1 (m) | 1,816.4 ± 256.2[‡] | 2,002.7 ± 134.6 | 2,006.0 ± 77.2[‡] | | 4.605 | 0.023 | 0.218 |

**Notes:**
SLJ, standing long jump; 5 × 25 m RSA, 5 × 25 m repeated sprint ability; YYIR1, Yo-Yo intermittent recovery test level 1.
[†] Significant difference between 13-year-old and 14-year-old players: $p < 0.05$.
[‡] Significant difference between 13-year-old and 15-year-old players: $p < 0.05$.
[§] Significant difference between 14-year-old and 15-year-old players: $p < 0.05$.

 

significantly. The height (ES = 0.468, $p < 0.001$), body mass (ES = 0.393, $p < 0.001$), thigh circumference (ES = 0.133, $p = 0.001$), 5 × 25-m repeated sprint speed (ES = 0.312, $p < 0.001$), $T$-test left (ES = 0.156, $p < 0.001$), $T$-test right (ES = 0.16, $p < 0.001$), standing long jump (ES = 0.456, $p < 0.001$), and 30-m sprint speed (ES = 0.198, $p < 0.001$) of 13–15-year-old players revealed that these indices of players with average maturation increased significantly; however, % Fat and 10-m sprint speed did not increase significantly. The height (ES = 0.249, $p = 0.009$), body mass (ES = 0.206, $p = 0.023$), thigh circumference (ES = 0.421, $p < 0.001$), 5 × 25-m repeated sprint speed (ES = 0.325, $p = 0.002$), standing long jump (ES = 0.451, $p < 0.001$), 10-m sprint (ES = 0.332, $p = 0.001$), 30-m sprint speed (ES = 0.538, $p < 0.001$), and YYIR1 (ES = 0.218, $p = 0.023$) of 13–15-year-old players revealed that these indices with late maturation increased significantly; however, % Fat and $T$-test (left and right) were not increased.

Table 3 depicts the correlations between body size (height, body mass, thigh circumference, and % fat) and physical capacities ($T$-test, standing long jump, 10-m and 30-m sprints, 5 × 25-m repeated sprints, and YYIR1). The height ($r = 0.619$, $p < 0.001$ and $r = 0.546$, $p < 0.001$, respectively) and body mass ($r = 0.524$, $p < 0.001$ and $r = 0.586$, $p < 0.001$, respectively) of 13–14-year-old players exhibited a significant correlation with standing long jump; the height of 13-year-old players manifested a significant correlation with the 30-m sprints ($r = -0.552$, $p < 0.001$) and 5 × 25-m repeated sprints ($r = -0.509$, $p < 0.001$); and the body mass of 15-year-old players displayed a significant correlation with 5 × 25-m RSA ($r = -0.548$, $p < 0.001$).

We show the slope between the CA and the physical characteristics of early-, average-, and late-maturing players aged 13–15 years in Table 4. All physical measures increased with CA, except for percent body fat. Biological maturity did not significantly influence fat content ($p > 0.05$) or the 10-m sprint ($p > 0.05$). The height ($r = 5.31$, $p < 0.001$; $r = 4.886$, $p < 0.001$), weight ($r = 4.52$, $p < 0.001$; $r = 4.265$, $p < 0.001$), standing long jump ($r = 11.794$, $p < 0.001$; $r = 13.576$, $p < 0.001$), and YYIR1 ($r = 83.418$, $p < 0.001$; $r = 84.623$, $p < 0.001$) slopes of the late- and average-maturing players were all greater than those of the early-maturing players, respectively. Late-maturing players showed greater thigh circumference ($r = 3.023$, $p = 0.001$), T left ($r = -0.346$, $p = 0.007$), R right ($r = -0.348$, $p = 0.004$), and 30-m sprint ($r = -0.2173$, $p < 0.001$) slopes relative to average- and early-maturing players.

## DISCUSSION

The purpose of this investigation was to characterize the influence of biological maturity (SA) on the physical and performance characteristics as well as growth and development of 13- to 15-year-old male football players. Our findings revealed that the physical measurements (height, body mass, and thigh circumference) of the 13–15-year-old late-maturing group were much lower than those of the early- and average-maturing groups. We also demonstrated that the $T$-test, 5 × 25-m RSA, and standing long jump of the early-maturing group surpassed the same indicators in the late-maturing group. The physical (except for % Fat) and performance characteristics of the 13–15-year-old players advanced significantly with age. There was no significant development in body fat

**Table 3 Correlation between physical measurements and physiological capacities[*].** Pearson correlation analysis of every physical measurement and physiological ability of players aged 13–15 years.

| 13 (y) | | 5 × 25 RSA | Left T | Right T | SLJ | 10 m | 30 m | YYIR1 |
|---|---|---|---|---|---|---|---|---|
| Height | r | −0.509[‡] | −0.429 | −0.369 | 0.619[‡] | −0.266 | −0.552[‡] | 0.215 |
| | p | <0.001 | 0.001 | 0.004 | <0.001 | 0.042 | <0.001 | 0.103 |
| Body mass (kg) | r | −0.385 | −0.391 | −0.339 | 0.524[‡] | −0.353 | −0.447 | 0.178 |
| | p | 0.003 | 0.002 | 0.009 | <0.001 | 0.006 | <0.001 | 0.177 |
| % Fat (%) | r | 0.374 | 0.234 | 0.210 | −0.235 | −0.298 | 0.007 | 0.206 |
| | p | 0.004 | 0.074 | 0.110 | 0.074 | 0.022 | 0.960 | 0.117 |
| C thigh | r | −0.419 | −0.334 | −0.326 | 0.491 | −0.319 | −0.444 | 0.265 |
| | p | 0.001 | 0.012 | 0.012 | <0.001 | 0.014 | <0.001 | 0.043 |
| 14 (y) | | | | | | | | |
| Height | r | −0.062 | 0.058 | 0.066 | 0.546[‡] | −0.107 | −0.235 | −0.207 |
| | p | 0.652 | 0.669 | 0.630 | <0.001 | 0.433 | 0.081 | 0.125 |
| Body mass (kg) | r | −0.217 | −0.064 | −0.053 | 0.586[‡] | −0.084 | −0.244 | −0.075 |
| | p | 0.107 | 0.641 | 0.699 | <0.001 | 0.538 | 0.070 | 0.582 |
| % Fat (%) | r | −0.070 | 0.003 | 0.011 | 0.309 | −0.122 | −0.210 | 0.009 |
| | p | 0.607 | 0.983 | 0.939 | 0.020 | 0.371 | 0.121 | 0.947 |
| C thigh | r | 0.008 | 0.026 | 0.047 | 0.308 | −0.303 | −0.366 | −0.276 |
| | p | 0.952 | 0.852 | 0.730 | 0.021 | 0.023 | 0.006 | 0.039 |
| 15 (y) | | | | | | | | |
| Height | r | −0.464 | −0.469 | −0.433 | 0.215 | −0.030 | −0.202 | 0.096 |
| | p | 0.001 | <0.001 | 0.001 | 0.125 | 0.832 | 0.151 | 0.496 |
| Body mass (kg) | r | −0.548[‡] | −0.402 | −0.364 | 0.310 | −0.166 | −0.227 | 0.130 |
| | p | <0.001 | 0.003 | 0.008 | 0.025 | 0.239 | 0.106 | 0.351 |
| % Fat (%) | r | 0.358 | 0.259 | 0.224 | 0.077 | −0.201 | −0.146 | 0.132 |
| | p | 0.009 | 0.064 | 0.110 | 0.587 | 0.154 | 0.301 | 0.352 |
| C thigh | r | 0.376 | −0.185 | −0.225 | 0.293 | 0.067 | −0.010 | −0.061 |
| | p | 0.006 | 0.190 | 0.109 | 0.035 | 0.636 | 0.944 | 0.666 |

Notes:
[*] The correlation coefficient was less than 1% and significant by two-tailed test.
[‡] $r > 0.5$.

**Table 4 Slope between CA and physical indicators in early-, average-, and late-maturing players aged 13–15 years.** The Slope of CA and physical indicators in early-, average-, and late-maturing players was calculated by least square linear regression equation.

| | | Height | Body mass | C thigh | % Fat | 5 × 25 RSA | Left T | Right T | SLJ | 10 m | 30 m | YYIR1 |
|---|---|---|---|---|---|---|---|---|---|---|---|---|
| CA | r | 4.8 | 4.71 | 2.275 | −0.017 | −1.158 | −0.26 | −0.274 | 12.992 | −0.043 | −0.152 | 84.036 |
| | p | <0.001 | <0.001 | <0.001 | 0.904 | <0.001 | <0.001 | <0.001 | <0.001 | 0.01 | <0.001 | <0.001 |
| Early | r | 1.823 | 4.107 | 1.943 | 0.382 | −0.886 | −0.201 | −0.234 | 6.767 | −0.026 | −0.042 | 71.642 |
| | p | 0.038 | 0.001 | 0.004 | 0.141 | <0.001 | 0.003 | 0.002 | 0.008 | 0.41 | 0.269 | 0.013 |
| Average | r | 4.886 | 4.265 | 1.825 | −0.11 | −1.058 | −0.216 | −0.229 | 13.576 | −0.03 | −0.143 | 84.623 |
| | p | <0.001 | <0.001 | 0.001 | 0.559 | <0.001 | <0.001 | <0.001 | <0.001 | 0.177 | <0.001 | <0.001 |
| Late | r | 5.31 | 4.52 | 3.023 | −0.38 | −1.361 | −0.346 | −0.348 | 11.794 | −0.074 | −0.217 | 83.418 |
| | p | <0.001 | <0.001 | 0.001 | 0.278 | <0.001 | 0.007 | 0.004 | <0.001 | 0.062 | <0.001 | 0.035 |

or 10-m sprinting ability for early-, average-, and late-maturing players; or for the 30-m sprint of early-maturing players. The height, standing long jumping, and 30-m sprint growth and development of average- and late-maturing players were also better than those of early-maturing players. Strength and conditioning coaches thus need to consider the maturity status of 13–15-year-old male football players when evaluating physical testing data, and to also recognize the nonlinear development of physical qualities.

We found that the 13–15-year-old elite male players who matured early were significantly taller and heavier than those who matured late; and that compared with the late-maturing group, the thigh circumferences of players 13 years of age and showing early and average maturity were significantly greater. Similarly, many studies on height and body mass have revealed that those adolescents who matured early showed better physiques than those who matured late (*Altimari et al., 2018*; *Itoh & Hirose, 2020*; *Teixeira et al., 2018*; *Figueiredo et al., 2009*).

In addition, the height and body mass of 13-year-old early-maturing players were also significantly elevated relative to those of average players, while there was no significant difference in height or body mass between early- and average-maturing players at 15 years of age. This finding suggests that early-maturing players begin to grow rapidly before the age of 13, which is consistent with Itoh's study of Japanese players (*Itoh & Hirose, 2020*).

The height, body mass, and thigh circumference of 13–15-year-olds who were players of average maturity grew and developed the most quickly, followed by late-maturing players (Table 2). The height growth of average- and late-maturing players was also significantly greater than that of early-maturing players. Although the thigh circumferences of 13–15-year-old players increased significantly, late-maturing players had the largest increase (ES = 0.421, $p < 0.001$; $r = 3.023$, $p = 0.001$). The body mass of early-, average-, and late-maturing players aged 13–15-years was also augmented significantly, while the percentage of body fat did not rise (Table 4).

The 13–15-year-old elite male football players within the early- and average-maturing groups were significantly superior to the late-maturing group in the standing long jump, which may be because the players' physical characteristics (height, body mass, thigh circumference) may have exerted a significant influence on their physical capacities. We found that there was a very significant positive correlation between the height and body mass of 13–14-year-old players and the standing long jump (Table 3). Players between the ages of 13 and 15 are in their pubertal period, where body-mass gain is primarily due to muscle growth (*Grendstad et al., 2020*), and the larger the muscle mass, the greater the explosive performance of the player.

In addition, the height and thigh circumference also positively influenced the explosiveness of youth footballers. The Sharma group (*Sharma et al., 2017*) reported that there was a very high positive correlation between height and vertical jump of top-ranking hockey players and cyclists; and others have shown a positive correlation between the cross-sectional area of a muscle and muscle strength and power. Further, improvements in muscle strength were associated with muscle hypertrophy (*Homma et al., 2019*; *Schlaeger et al., 2019*). The height slopes for average- and late-maturing players were

also greater than for early-maturing players (Table 4). Thus, our study suggests that the elevation in height was an important reason for the explosive growth of 13- to 15-year-old average- and late-maturing players, and that the explosive growth of late-maturing players was related to the enlargement in thigh circumference.

The *T*-test times of early-maturing players were significantly better than those of late-maturing players, and the *T*-test performances of 13–15-year-old early-, average-, and late-maturing players all improved greatly. Agility is principally related to the speed of directional change, perception, and decision-making factors (*Young, James & Montgomery, 2002*); and neuromuscular system training can improve sensitivity (*DiStefano et al., 2010*). However, *Itoh & Hirose (2020)* reported that there were no significant differences among the early-, average- or late-maturing groups in agility with respect to the 10-m*5 repeated sprints, which may be related to the different evaluation methods used in assessing agility.

The influence of biological maturity on the 30-m sprints and $5 \times 25$-m repeated sprints diminished with age. Thirteen-year-old early-maturing players performed significantly better on 30-m sprints than average- and late-maturing players, and the average-maturing players were also significantly better than the late-maturing players. However, there was no significant difference in 30-m sprint results with respect to biological maturity at 15 years of age (Table 2). We noted a significant negative correlation between height and the 30-m sprint (Table 3), which was consistent with the reports by *Katoh et al. (1999)* and *Itoh & Hirose (2020)*. *Itoh & Hirose (2020)* also reported that the growth of adolescent players' strides was an important factor influencing running speed. Thus, we believe that the increase in height of late-maturing players is an important reason for the observed advancements in speed.

The ability to repeatedly perform high-intensity activities is crucial to players' performance (*Dodd & Newans, 2018*). At 13 years old, the early-maturing players' $5 \times 25$-m repeated sprints were significantly better than the average- and late-maturing players; and players of average maturity were also better than the late-maturing players. However, we observed no significant difference in the $5 \times 25$-m repeated sprints among the early-, average-, and late-maturing players at 15 years of age. There were significant correlations between height, body mass, % Fat, and thigh circumference of the 13- and 15-year-old players and the $5 \times 25$-m repeated sprints (Table 3). This observation is commensurate with the biological maturity–related changes in body size, strength, power, and speed of youth football players (*Figueiredo et al., 2009*; *Duarte et al., 2019*). The $5 \times 25$-m RSA of 13–15-year-old early-, average-, and late-maturing players also increased significantly; and of these groups, the late-maturing players demonstrated the most significant growth (Table 4). *Valente-Dos-Santos et al. (2012)* posited that the repeated-sprint ability of youth football players was affected by CA, SA, body fat, and limb explosive power; and our study results are congruent with this hypothesis.

Although the 10-m sprint time of 13-year-old players of early- and average maturation was significantly faster than that of late-maturing players, biological maturity did not manifest a significant effect on improving the 10-m sprint of 13–15-year-old players. As previously discussed, this aspect may be related to the height, body mass, and thigh

circumference of early- and average-maturing players. In addition, the 10-m sprint may also be influenced by relative strength, running mechanics, and neuromuscular control (*Meyers et al., 2015*). We therefore infer that biological maturity does not have a significant influence on the 10-m sprint ability of players aged 13–15 years, but does increase with CA.

There were no significant differences in aerobic endurance (YYIR1) among the early-, average- and late-maturing players aged 13–15 years; nor was there a significant correlation with height, body mass, % Fat, or thigh circumference; these data are consistent with numerous previous studies (*Peña-González et al., 2018*; *Roescher et al., 2010*; *Matta et al., 2015*). The present study suggests that the anthropometric characteristics exerted no significant action on intermittent endurance, which agrees with the work by *Roescher et al. (2010)* In addition, we posit that both developmental maturity and CA engender a significant effect on the improvement in intermittent- endurance ability of football players aged 13–15 years.

This study possesses several limitations. First, although the 30-m sprint, 5 × 25-m RSA, and standing long jump are biologically dependent upon maturity, we were unable to clarify which of these were influenced by anthropometric variables as we only showed their correlations and could not show causality. Second, we were unable to obtain information on the number of training years experienced by the players, which may have influenced their physical performance. Finally, we adopted a cross-sectional design, and thus recommend that future investigators employ longitudinal designs to infer developmental trajectories as opposed to solely assessing differences by maturational status.

## CONCLUSIONS

In conclusion, our study results confirmed that the anthropometics (height and body mass) and physical characteristics (5 × 25-m RSA, standing long jump, and *T*-tests) of 13- to 15-year-old early-maturing players were significantly superior to the late-maturing players. The influence of biological maturity on physical characteristics (except for % Fat) of players aged 13–15 decreased with age and did not influence YYIR1. Height, 30-m sprint, standing long jump, and YYIR1 growth of late- and average-maturing players were greater than for early-maturing players. Given that the 30-m and 5 × 25-m repeated sprint times and standing long jump distances were shown to exhibit significant correlations with physical characteristics, these physical indices may have been affected by anthropometric variables such as height and body mass due to biological maturity.

### Funding
The authors received no funding for this work.

### Competing Interests
The authors declare that they have no competing interests.

## Author Contributions

- Shidong Yang conceived and designed the experiments, performed the experiments, analyzed the data, prepared figures and/or tables, and approved the final draft.
- Haichun Chen conceived and designed the experiments, performed the experiments, authored or reviewed drafts of the paper, and approved the final draft.

## Ethics

The following information was supplied relating to ethical approvals (*i.e.*, approving body and any reference numbers):

Fujian Normal University "Ethics Review Procedures Concerning Research with Human Subjects".

## Data Availability

The raw measurements are available in the Supplemental File.

## Supplemental Information

Supplemental information for this article can be found online at http://dx.doi.org/10.7717/peerj.13282#supplemental-information.

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
