# Peer review of "Physical characteristics of elite youth male football players aged 13–15 are based upon biological maturity"

_PeerJ, doi:10.7717/peerj.13282_

## Round 0.1 · original submission · Major Revisions

Dear Authors,

Two experts in the fields revised your manuscripts and identified several major points you should consider in the revision process.

Reviewer 1 ·

Basic reporting

Comments to the editor
Dear Editor,
I consider this article interesting and sufficiently appropriate in providing additional and extended knowledge on the role of maturity to discriminate the level of play in young football players. However, the authors are invited to proofread the English language of the text to make it clearer and more convincing. Moreover, the authors should reinforce the content of the introduction to improve the rationale. Other changes are required to improve the manuscript. Here below additional specific comments

Intro
Line 39: suggested rewording: “Studies revealed that … are superior in anthropometrical (height, body mass, and body fat) [cite here] and performance (speed, agility…) [cite here] variables.”
Line 47: “replace “abilities” to “characteristics”. Please replace it throughout the manuscript.
Line 50: “football clubs” instead of “blub teams”
Line 53: a reference is needed.
Lines 56-57: please rephrase for clarity.
Line 57: please describe “TW”.
Line 61: “correlation with them”. Remove “between physical and physiological abilities”.
Line 62: fix “strentht”
Line 62: other?? Please be specific
Line 67: what about study hypothesis?
Inside the introduction, the Authors are encouraged to reinforce the rationale by also implementing other information from more recent papers. Especially for the use of anthropometrical (both body dimension and body composition data) variables.
The authors are invited to consider the following articles to address this point:
1) Bioimpedance Vector References Need to Be Period-Specific for Assessing Body Composition and Cellular Health in Elite Soccer Players: A Brief Report
https://doi.org/10.3390/jfmk5040073

2) Importance of anthropometric features to predict physical performance in elite youth soccer: a machine learning approach
doi: 10.1080/15438627.2020.1809410
3) Anthropometric and Functional Profile of Selected vs. Non-Selected 13-to-17-Year-Old Soccer Players
doi: 10.3390/sports8080111

Methods

Line 70: suggested rewording: “were performed during the fourth weekend of September and during the first and second week of October prior to start…”
Line 76: “All subjects…” this sentence is redundant. Please remove it.
Line 82: please fix the refusal
Line 85: please DO NOT use “physiological ability” to express a physical capacity or quality coming from a (physical) performance tests. Use sprint performance, or agility/change of direction, or aerobic performance.
Line 119: “to obtain” instead of “take”
Lines 154-155: please rephrase
Line 161: please fix “Fatand”
Please DO NOT use “p = 0.000”. You can replace it as p < 0.001 or p < 0.0001 if you have four 0 in a row.
Discussion
An English language revision is needed to raise the quality of the discussion.
Line 190: “significantly lower” appears misleading when linked with sprint performance. Perhaps, it would be more informative, replacing it as “better performance” or similar.
Line 191: reformulate “physiological ability” as suggested in a previous comment. Please apply this correction throughout the manuscript.
Lines 192-193: please rephrase for clarity.

Table 1: please fix/adjust the columns (F and p-value) width. Please also explicit the “F”, “p”, and “n2” columns. I understand they are common statistical parameters. However, for the sake of clarity, it would be appropriate to report them clearly.

Experimental design

no comment

Validity of the findings

no comment

Additional comments

Please, an English language revision in needed within the entire manuscript.

Reviewer 2 ·

Basic reporting

Dear Authors,
I would like to encourage you to improve the English style of your article. Moreover there are some few error in the text. Please, revise them. The Author should revise dramatically their results section. Please, see below.

Experimental design

I believe that the results are not so original and quite weak. It is not enough to conclude that players in early maturity stage have better anthropometric and physical performance characteristics. That's clear and well know. To provide this in a Chinese population is not enough to justify this work.

I suggest to the Author to revise their result section and, in turn, discussion. I believe the Authors should remove or reduce actual results and provide correlations between Skeletal age and physical output during test. You are aiming to assess "the relationship between biological maturity and physical characteristics. You may correlated these factors, while you are showing us only the relationship between physical measurements and physiological capacities (Table 2). Moreover, you are describing the characteristics of different groups (Table 1 and table 3). Conversely, you have to correlate SA and CA with different physical outputs. When you'll have the slope between SA or CA with physical results for each test, you should provide a comparison between the slope of the relationship between maturity status vs physical results using chronological age or Scheletal age.

Differences in these slope will help you to provide and clarify the effects of scheletal maturity on test, suggesting that SA approach describe better the maturity vs test relationship that CA approach. These may be provide for each test you utilised.

After that the discussion should be completely change. I'll look at that after these changes.

Validity of the findings

Please see above comments

Additional comments

Line 38 to 45 should be reworded. "physically, body mass, and body fat and physiologically...." this phrase should be grammatically improved and corrected.

Line 49-51. Already reported in the first section, please removed.

Line 56 to 57: please, provide more REFs.

Line 58: What is TW3, please clarify across the introduction.

Line 60-61: not clear, please remove or revise it.

Line 66: Why is important to abbreviate S&C?

Line 71: please, remove injury information because of this information is already reported in “subjects” section.

Line 97: "motor" measurements ???

Please, provide a clear review of the literature about the relationship between SA and physical tests in different population (what test other performed? What results?). The use of more papers clearly based on this aspect will be appreciated. Additionally, clarify why you study add new findings that previous one, please.

Annotated reviews are not available for download in order to protect the identity of reviewers who chose to remain anonymous.

---

## Round 0.2 · Minor Revisions

Dear Authors, tone of the reviewers revised your R1 manuscript version and found some minor issues you should consider in the next version.

Reviewer 1 ·

Basic reporting

The authors are invited to check the text format throughout the manuscript.

see lines 39-47

Experimental design

no comment

Validity of the findings

no comment

Additional comments

Abstract
I would suggest the authors to reduce the text linked to the Results section.

Statistical analysis

line 182: be specific on the use of eta squared or Cohen's d, which belong to two different standardized family of effect size.

---

## Round 0.3 · accepted · Accept

Dear Author, I have personally checked your revised manuscript and, in my opinion, you have satisfactorily replied to all the issues retrieved by the reviewers. I think your manuscript could be in an acceptable form now.
Best regards.